# Portfolio Optimization of Photovoltaic/Battery Energy Storage/Electric Vehicle Charging Stations with Sustainability Perspective Based on Cumulative Prospect Theory and MOPSO

**Jicheng Liu [1,2] and Qiongjie Dai [1,2,3,*]**

1    School of Economics and Management, North China Electric Power University, Changping, Beijing 102206, China; ljch@ncepu.edu.cn

2    Beijing Key Laboratory of New Energy and Low-Carbon Development, North China Electric Power University, Changping, Beijing 102206, China

3    School of Mathematics and Computer Engineering, Ordos Institute of Technology, Ordos 017000, China

*    Correspondence: daiqiongjie@yeah.net; Tel.: +86-187-4775-3183

**Abstract:** Recently, an increasing number of photovoltaic/battery energy storage/electric vehicle charging stations (PBES) have been established in many cities around the world. This paper proposes a PBES portfolio optimization model with a sustainability perspective. First, various decision-making criteria are identified from perspectives of economy, society, and environment. Secondly, the performance of alternatives with respect to each criterion is evaluated in the form of trapezoidal intuitionistic fuzzy numbers (TrIFN). Thirdly, the alternatives are ranked based on cumulative prospect theory. Then, a multi-objective optimization model is built and solved by multi-objective particle swarm optimization (MOPSO) algorithm to determine the optimal PBES portfolio. Finally, a case in South China is studied and a scenario analysis is conducted to verify the effectiveness of the proposed model.

**Keywords:** portfolio optimization; electric vehicle charging station; photovoltaic; energy storage; sustainability

## 1. Introduction

In recent years, with the rapid increase of electric vehicles (EV), a lot of EV charging stations are built but the problem of high pressure on the power grid cannot be ignored. Consequently, a solution of small-scale photovoltaic (PV)/battery energy storage/EV charging station (PBES) is proposed. In this system, the electricity is generated by PV modules and the batteries can adjust the balance of energy supply and demand. Therefore, PBES can be considered as a microgrid that is more flexible, environmentally friendly, and economical. Besides, PBES can exchange electricity with a utility grid if needed, which relieves more stress of the charging load than the normal EV charging stations. Recently, a number of pilot PBES projects are carried out. For example, some PBES programs have been implemented in many cities of China, such as Qingdao, Hainan, Shanghai, etc.

Recently, many studies investigated the feasibility and structure of EV charging stations with local renewable energy supply. For instance, Karmaker et al. [1] connected PV modules and biogas generators with EV charging stations in Bangladesh based on local resource distribution. However, in most situations, PV energy is more easily accessible because PV modules can be installed on flat rooftops. Ye et al. [2] proposed a solar-powered EV charging station model and studied its economic feasibility. Ul-Haq et al. [3] proposed a smart grid-connected EV charging station architecture supplied

by PV generation. Chandra Mouli et al. [4] designed a solar-powered EV charging station in workplaces in the Netherlands and investigated its possibilities. Besides, some research has also considered the role of the energy storage system in the design of PV-powered EV charging stations. Savio et al. [5] analyzed the energy management strategy in a PV microgrid structured EV charging station where the PV power was stored in batteries when vehicles were not connected for charging. Esfandyari et al. [6] evaluated the technical performance of a PBES in a campus microgrid with the aim of maximizing self-consumption and autonomy. García-Triviño et al. [7] proposed a control strategy of a fast EV charging station consisting of a PV system, batteries, fast charging units, and a connection to the local grid. It can be concluded from the above research that PBES has attracted a lot of attention and proved to be a feasible solution to reduce the stress of EV charging load on the utility grid.

The research on PBES mostly focused on the technical design and energy management strategy. However, with the increase of PBES projects, the portfolio optimization problem is also a critical research topic because it is important for enterprises to make optimal investment decisions within limited costs or resources. The portfolio optimization is one of the main steps in project management and many studies are found in this field, such as new product project portfolio management [8] and research and development (R&D) project portfolio selection [9]. In addition, in the field of energy project portfolio optimization, some valuable research has been carried out. Wu et al. [10] built a fuzzy multi-criteria decision-making (MCDM) framework to optimize renewable energy project portfolio in an efficient way. Faia et al. [11] improved particle swarm optimization algorithm to solve the optimization model of electricity market players' participation portfolio. Zeng et al. [12] developed a multi-objective model of energy generation portfolios to minimize cost and risk. In this literature, we can see that more than one objective is taken into account and sustainability is a significant issue to be considered which is a trade-off among economic, social, and environmental activities [10].

The portfolio optimization problem includes two main steps: Evaluating the performance of projects and optimizing the project portfolio. In the existing research, MCDM methods and evolutionary algorithms (EA) are the most widely used techniques to solve the portfolio optimization problem. Hashemizadeh and Ju [13] combined the technique for order preference by similarity to ideal solution (TOPSIS) and geographic information system (GIS) technology to determine strategic-aligned projects portfolio. Wu et al. [10] used the analytic hierarchy process (AHP) and interval type-2 fuzzy weighted averaging operator to score renewable energy projects, and applied non-dominated sorting genetic algorithm II to select the optimal portfolio. Tavana et al. [14] proposed a three-stage hybrid method in which data envelopment analysis (DEA) and TOPSIS were employed to rank projects and integer programming was used to select the most suitable project portfolio. During the decision process, decision-makers always give comments under uncertain circumstances, and therefore, some vague and fuzzy information should be handled. From the above literature, it can be seen that the fuzzy set theory is an effective method to quantify the fuzzy information given by decision-makers. For example, Huang et al. [15] applied trapezoidal fuzzy number to measure the linguistic information and proposed a fuzzy AHP method to select government-sponsored R&D projects. Khalili-Damghani et al. [16] established a DEA and EA based framework to deal with sustainable project portfolio selection under fuzzy environment. Therefore, this study will employ trapezoidal intuitionistic fuzzy sets to deal with fuzzy information in the decision process.

Based on the existing research, this study proposes a hybrid decision framework under trapezoidal intuitionistic fuzzy environment to optimize the PBES portfolio from a sustainability perspective. In this method, the cumulative prospect theory is used to evaluate the performance of potential PBES projects and multi-objective particle swarm optimization (MOPSO) algorithm is applied to select the optimal portfolio. The main contributions of this study are presented as follows:

(1)  Considering the sustainable development of PBES, this paper identified 14 decision-making criteria from perspectives of economy, society, and environment to evaluate the performance of PBES.

(2) A hybrid decision framework to select the optimal PBES portfolio is proposed. Firstly, TrIFNs are utilized to handle linguistic terms. Then each potential project is evaluated by cumulative prospect theory. Finally, a multi-objective integer programming model is established to optimize the portfolio.

(3) A case in South China is studied, in which 10 potential PBES projects are evaluated from a sustainable perspective and the optimal portfolio is obtained. In addition, six scenarios are analyzed to prove the reliability of the result.

## 2. Decision-Making Criteria Identification

Considering the sustainable development of PBES, three objectives are selected and analyzed: Economic, social, and environmental values. From these perspectives, 14 criteria are identified to evaluate the investment value of PBES, illustrated in Figure 1. The specific description of each criterion is presented as follows.

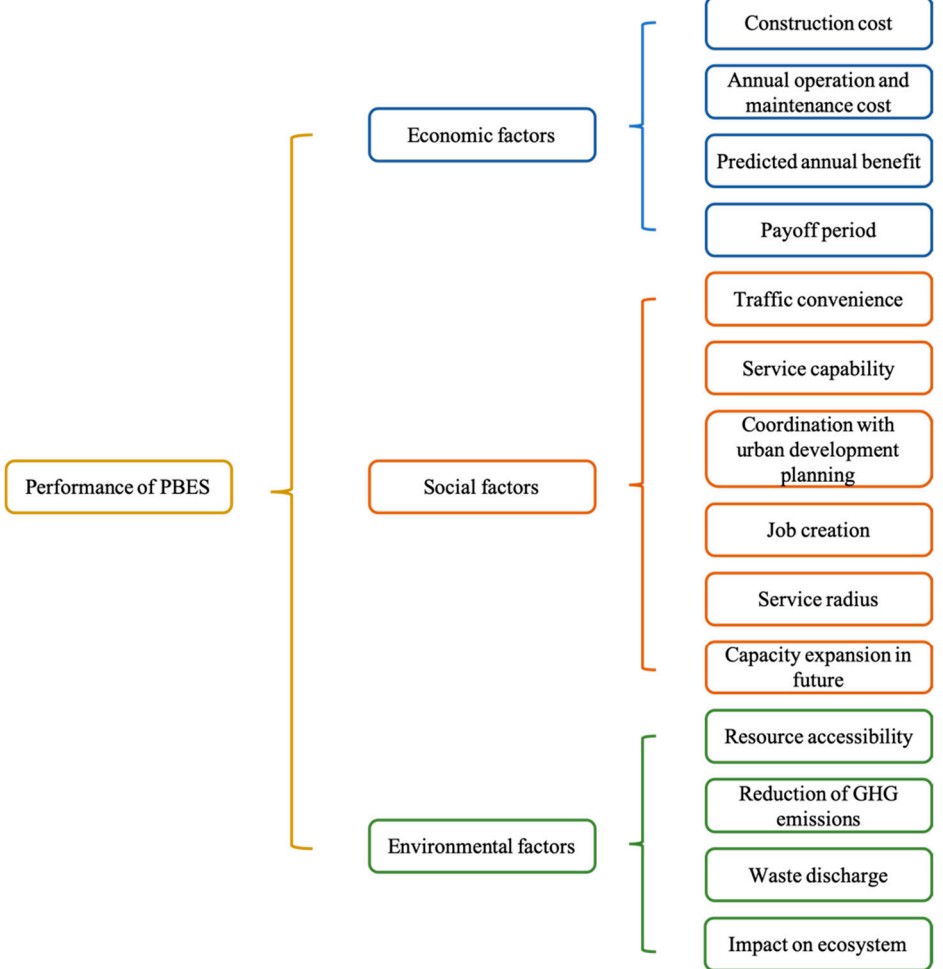

**Figure 1.** The decision-making criteria for photovoltaic/battery energy storage/electric vehicle charging stations (PBES) portfolio optimization.

### 2.1. Economic Factors

(1) Construction cost (C11) [17–20]: Includes land cost, charging facility cost, PV array cost, battery cost, installation cost, and other fixed costs.

(2) Annual operation and maintenance cost (C12) [17,18]: Includes staff wages, cost of purchasing power from the utility grid when PV power is insufficient, and maintenance cost of equipment, etc.

(3)  Predicted annual benefit (C13) [18]: Includes the profit gained from charging service and selling electricity to the utility grid. This criterion indicates the profitability of the targeted PBES.

(4)  Payoff period (C14) [21]: Represents the ability to recoup the investment, which is an important indicator for investors.

## *2.2. Social Factors*

(1)  Traffic convenience (C21) [17,18,22,23]: Refers to the condition of traffic flow, the number of main roads and the convenience of accessing public transport, etc. Convenient traffic can help drivers arrange their traveling more easily, which shows great potential to attract more customers.

(2)  Service capability (C22) [17,21]: Refers to the capability to serve EV customers, which can be evaluated by the number of fast and slow charging facilities, the expected opening hours, and the number of potential EV owners around the target area, etc.

(3)  Coordination with urban development planning (C23) [18]: Indicates the effect of targeted PBES on urban construction. If a PBES coordinates with urban traffic planning and utility grid planning, it will better undertake the responsibility of serving the society.

(4)  Job creation (C24) [10]: Refers to the ability to create job opportunities for the society, especially for local communities.

(5)  Service radius (C25) [22,24]: Measures the distance between the targeted PBES and the adjacent electric vehicle charging station. This criterion indicates the convenience degree for EV owners to get access to charging service.

(6)  Capacity expansion in the future (C26) [19]: Refers to the possibility to expand the scale of the targeted PBES. As the number of EV owners increases, PBES may need to provide more services.

## *2.3. Environmental Factors*

(1)  Resource accessibility (C31) [19,22,25]: Measures the ability to access the required resources such as land and solar energy, etc. The land requirement may influence some farmland or landscapes. In addition, easy access to abundant solar energy will make better use of energy resources and reduce the use of PV modules.

(2)  Reduction of greenhouse gas (GHG) emissions (C32) [1,20,21]: Measures the air pollutant reduction by constructing a PBES rather than a normal electric vehicle charging station. The PBES uses photovoltaic energy to provide electricity and sell excessive electricity to the utility grid, which is a cleaner way than other charging stations.

(3)  Waste discharge (C33) [23,24]: Refers to the discharge of sewage, construction garbage, and battery disposal, etc.

(4)  Impact on the ecosystem (C34) [10]: Measures the environmental friendliness of the targeted PBES. A better PBES should have less impact on the environment.

## 3. Basic theory and Methodology

In this study, a hybrid decision framework under intuitionistic fuzzy environment is proposed to investigate the optimal portfolio for PBES in the targeted area. In the proposed decision framework, some relevant methods, including trapezoidal intuitionistic fuzzy sets, cumulative prospect theory, and MOPSO algorithm, are utilized. The description of these theories is presented in this subsection.

## *3.1. Trapezoidal Intuitionistic Fuzzy Sets*

**Definition 1.** Let $X$ be the universe of discourse. Then an intuitionistic fuzzy set in $X$ is defined as Equation (1).

$$A = \{< X, \mu_A(x), v_A(x) | x \in X >\} \tag{1}$$

where $\mu_A(x) : X \rightarrow [0,1]$ and $v_A(x) : X \rightarrow [0,1]$. $\mu_A(x)$ and $v_A(x)$ are the membership degree and non-membership degree for $x \in X$, respectively, satisfying the condition that $0 \leq \mu_A(x) + v_A(x) \leq 1$. The hesitancy degree of for $x \in X$ is given by $\pi_A(x) = 1 - \mu_A(x) - v_A(x)$.

**Definition 2.** Let $\widetilde{a}$ be an intuitionistic fuzzy number on the set of real numbers whose membership degree and non-membership degree are expressed in Equation (2).

$$
\mu_{\widetilde{a}}(x) = \begin{cases} \frac{x-a}{b-a}\mu_{\widetilde{a}} & a \leq x < b \\ \mu_{\widetilde{a}} & b \leq x \leq c \\ \frac{d-x}{d-c}\mu_{\widetilde{a}} & c < x \leq d \\ 0 & x < a \text{ or } x > d \end{cases} \quad v_{\widetilde{a}}(x) = \begin{cases} \frac{b-x+(x-a')}{b-a'}v_{\widetilde{a}} & a' \leq x < b \\ v_{\widetilde{a}} & b \leq x \leq c \\ \frac{x-c+(d'-x)}{d'-c}v_{\widetilde{a}} & c < x \leq d' \\ 1 & x < a' \text{ or } x > d' \end{cases} \quad (2)
$$

where $a, a', b, c, d, d'$ are real numbers and $a' \leq a \leq b \leq c \leq d \leq d'$. $\mu_{\widetilde{a}}$ and $v_{\widetilde{a}}$ are the maximum of membership degree and the minimum of non-membership degree, respectively, with the condition that $0 \leq \mu_{\widetilde{a}} \leq 1$, $0 \leq v_{\widetilde{a}} \leq 1$ and $\mu_{\widetilde{a}} + v_{\widetilde{a}} \leq 1$. Then, $\widetilde{a}$ is called a TrIFN, denoted as $\widetilde{a} = \; <(a,b,c,d),(a',b,c,d'); \mu_{\widetilde{a}}, v_{\widetilde{a}}>$ (as shown in Figure 2). The hesitancy degree of $\widetilde{a}$ is given by $\pi_{\widetilde{a}} = 1 - \mu_{\widetilde{a}} - v_{\widetilde{a}}$ which represents the uncertainty of $\widetilde{a}$.

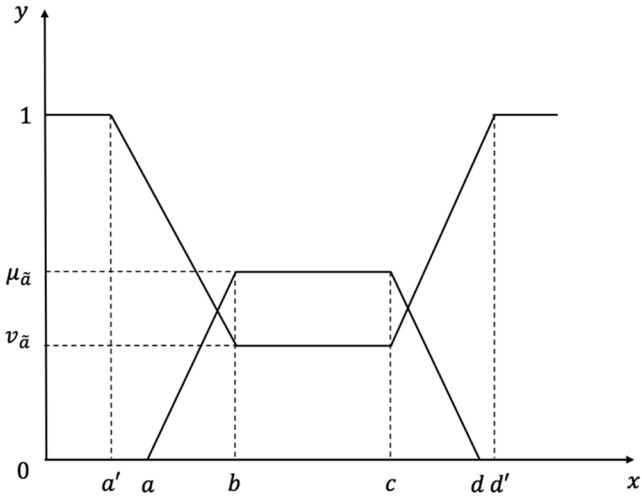

**Figure 2.** TrIFN.

If $a = a'$ and $d = d'$, $\widetilde{a}$ can be simplified as $\widetilde{a} = [(a,b,c,d); \mu_{\widetilde{a}}, v_{\widetilde{a}}]$. Obviously, when $b = c$, the TrIFN $\widetilde{a}$ degrades into a triangular intuitionistic fuzzy number (TIFN). Furthermore, if $\mu_{\widetilde{a}} = 1$ and $v_{\widetilde{a}} = 0$, then the TrIFN $\widetilde{a}$ is called a normal TrIFN.

**Definition 3.** Let $\widetilde{a_1} = \; <(a_1,b_1,c_1,d_1),(a'_1,b_1,c_1,d'_1); \mu_{\widetilde{a_1}}, v_{\widetilde{a_1}}>$ and $\widetilde{a_2} = \; <(a_2,b_2,c_2,d_2),(a'_2,b_2,c_2,d'_2); \mu_{\widetilde{a_2}}, v_{\widetilde{a_2}}>$ be two TrIFNs and $\lambda$ be any real number. Then the operational rules between two TrIFNs are defined as follows.

$$
\begin{aligned}
\widetilde{a_1} + \widetilde{a_2} = \; &<(a_1 + a_2, b_1 + b_2, c_1 + c_2, d_1 + d_2), (a'_1 + a'_2, b_1 + b_2, c_1 + c_2, d'_1 + d'_2); \\
&\min(\mu_{\widetilde{a_1}}, \mu_{\widetilde{a_2}}), \max(v_{\widetilde{a_1}}, v_{\widetilde{a_2}})>
\end{aligned} \quad (3)
$$

$$
\lambda\widetilde{a_1} = \; <(\lambda a_1, \lambda b_1, \lambda c_1, \lambda d_1), (\lambda a'_1, \lambda b_1, \lambda c_1, \lambda d'_1); \mu_{\widetilde{a_1}}, v_{\widetilde{a_1}}> \text{ if } \lambda > 0 \quad (4)
$$

$$
\lambda\widetilde{a_1} = \; <(\lambda a_1, \lambda c_1, \lambda b_1, \lambda d_1), (\lambda a'_1, \lambda c_1, \lambda b_1, \lambda d'_1); \mu_{\widetilde{a_1}}, v_{\widetilde{a_1}}> \text{ if } \lambda < 0 \quad (5)
$$

$$
\widetilde{a_1} \cdot \widetilde{a_2} = \; <(a_1 a_2, b_1 b_2, c_1 c_2, d_1 d_2), (a'_1 a'_2, b_1 b_2, c_1 c_2, d'_1 d'_2); \min(\mu_{\widetilde{a_1}}, \mu_{\widetilde{a_2}}), \max(v_{\widetilde{a_1}}, v_{\widetilde{a_2}})> \quad (6)
$$

**Definition 4** [26].　　　　Let $\widetilde{a_1} = \; <(a_1, b_1, c_1, d_1), (a'_1, b_1, c_1, d'_1); \mu_{\widetilde{a_1}}, v_{\widetilde{a_1}}>$　and　$\widetilde{a_2} =$ $<(a_2, b_2, c_2, d_2), (a'_2, b_2, c_2, d'_2); \mu_{\widetilde{a_2}}, v_{\widetilde{a_2}}>$ be two TrIFNs.　Then the hamming distance between $\widetilde{a_1}$ and $\widetilde{a_2}$ is defined by Equation (7).

$$
\begin{aligned}
d_H(\widetilde{a_1}, \widetilde{a_2}) = 1/8 \big[ & \left| \mu_{\widetilde{a_1}} a_1 - \mu_{\widetilde{a_2}} a_2 \right| + \left| \mu_{\widetilde{a_1}} b_1 - \mu_{\widetilde{a_2}} b_2 \right| + \left| \mu_{\widetilde{a_1}} c_1 - \mu_{\widetilde{a_2}} c_2 \right| + \left| \mu_{\widetilde{a_1}} d_1 - \mu_{\widetilde{a_2}} d_2 \right| + \\
& \left| (1 - v_{\widetilde{a_1}}) a'_1 - (1 - v_{\widetilde{a_2}}) a'_2 \right| + \left| (1 - v_{\widetilde{a_1}}) b_1 - (1 - v_{\widetilde{a_2}}) b_2 \right| + \left| (1 - v_{\widetilde{a_1}}) c_1 - (1 - v_{\widetilde{a_2}}) c_2 \right| \\
& + \left| (1 - v_{\widetilde{a_1}}) d'_1 - (1 - v_{\widetilde{a_2}}) d'_2 \right| \big]
\end{aligned}
\tag{7}
$$

**Definition 5** [26]. Let $\widetilde{a} = \;<(a, b, c, d), (a', b, c, d'); \mu_{\widetilde{a}}, v_{\widetilde{a}}>$ be a TrIFN. Then the defuzzified value of $\widetilde{a}$ is given by Equation (8).

$$
D(\widetilde{a}) = \frac{1}{12} [(a + b + c + d)\mu_{\widetilde{a}} + (a' + b + c + d')(1 - v_{\widetilde{a}})]
\tag{8}
$$

*3.2. Cumulative Prospect Theory*

In real situations, in addition to the fuzzy decision information, decision-makers often have a certain risk preference and psychological behavior characteristics as a result of different experience and knowledge structure. Therefore, Kahneman and Tversky proposed cumulative prospect theory to consider the subjective preferences of decision-makers in order to avoid the influence of the subjective judgment on decision results. In cumulative prospect theory, the prospect value *V* is obtained by the value function $v(x)$ and the weight function $\pi$, calculated by Equation (9).

$$
V(x) = \sum_{i=1}^{n} \pi(\omega_i) v(x_i)
\tag{9}
$$

The value function represents the risk preference and is defined by Equation (10). The schematic diagram of the value function is illustrated in Figure 3.

$$
v(x) = \begin{cases} x^{\alpha} & x \geq 0 \\ -\theta(-x)^{\beta} & x < 0 \end{cases}
\tag{10}
$$

where $x \geq 0$ and $x < 0$ represent gains and losses, respectively. $\alpha, \beta$ are exponential parameters which represent the concavity and convexity of the value curve. $\lambda$ is the risk aversion parameter and $\lambda > 1$ denotes loss aversion. Here, $\alpha = \beta = 0.88$ and $\theta = 2.55$ are adopted based on Kahneman and Tversky's research [27].

The weight function means the decision weights and is given by Equation (11).

$$
\pi(\omega_j) = \begin{cases} \pi^+(\omega_j) = \dfrac{\omega_j^{\gamma^+}}{[\omega_j^{\gamma^+} + (1 - \omega_j)^{\gamma^+}]^{\frac{1}{\gamma^+}}} & x \geq 0 \\[3mm] \pi^-(\omega_j) = \dfrac{\omega_j^{\gamma^-}}{[\omega_j^{\gamma^-} + (1 - \omega_j)^{\gamma^-}]^{\frac{1}{\gamma^-}}} & x < 0 \end{cases}
\tag{11}
$$

where the coefficients $\gamma^+ = 0.61$ and $\gamma^- = 0.69$.

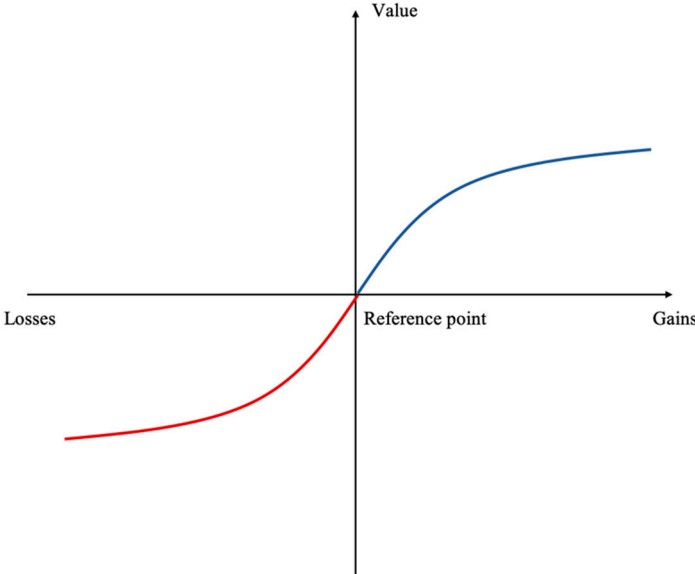

**Figure 3.** The schematic diagram of value function.

### 3.3. Multi-Objective Particle Swarm Optimization Algorithm

Particle swarm optimization (PSO) algorithm was proposed by Kennedy and Eberhart in 1995, which is based on the research on social behavior by birds. In this algorithm, each particle has its velocity and position, which are updated during iterations based on Equations (12) and (13). At each iteration, it is significant to determine the local and global best solutions of particles in order to achieve the global optimum eventually.

$$V_i(t) = \omega V_i(t-1) + c_1 r_1(pbest_i(t-1) - p_i(t-1)) + c_2 r_2(gbest_i(t-1) - p_i(t-1)) \tag{12}$$

$$p_i(t) = p_i(t-1) + V_i(t) \tag{13}$$

In the above equations, $c_1$, $c_2$ are learning factors, $r_1$, $r_2$ are random numbers between 0 and 1, $pbest_i$ and $gbest_i$ are the personal and global best solution respectively, and $\omega$ is the inertia weight, which can be calculated as follows.

$$\omega = \omega_{max} - (\omega_{max} - \omega_{min}) \times \frac{t}{t_{max}} \tag{14}$$

where $\omega_{max}$ and $\omega_{min}$ are the maximum and minimum of inertia weight, $t$ is the current iteration, and $t_{max}$ is the maximum number of iterations.

In multi-objective optimization problems, there are several non-dominated solution sets in the population. In this study, the crowding distance, which represents the distance between adjacent elements is employed to rank non-dominated solutions in the same level, defined by Equation (15). During the selection of leader particles, a greater crowding distance means a higher opportunity of selection. In this way, the global best solution can be determined even though the non-dominated solution set comprises several leaders.

$$CD_i = \frac{\left|f_1(x_{i+1}) - f_1(x_{i-1})\right|}{f_1^{max} - f_1^{min}} + \frac{\left|f_2(x_{i+1}) - f_2(x_{i-1})\right|}{f_2^{max} - f_2^{min}} + \frac{\left|f_3(x_{i+1}) - f_3(x_{i-1})\right|}{f_3^{max} - f_3^{min}} \tag{15}$$

The structure of the proposed MOPSO is presented in Figure 4.

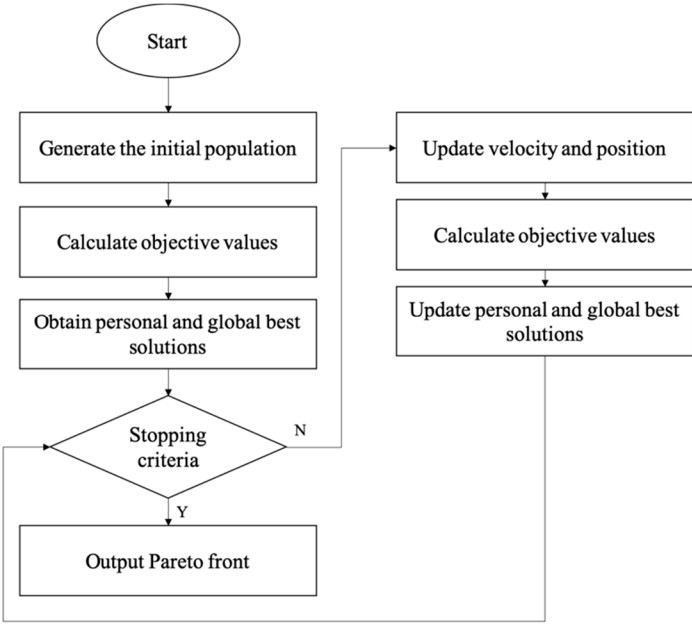

**Figure 4.** The structure of the proposed multi-objective particle swarm optimization (MOPSO).

## 4. Framework for Portfolio Optimization of PBES

To begin with, the objective values of the candidate PBES projects are determined based on cumulative prospect theory in which TrIFNs are employed to describe fuzzy information during the decision-making process. Then a multi-objective integer programming model is established to determine the optimal portfolio of PBES considering economic, social, and environmental objectives. The detailed process of the proposed decision framework includes three phases, illustrated in Figure 5.

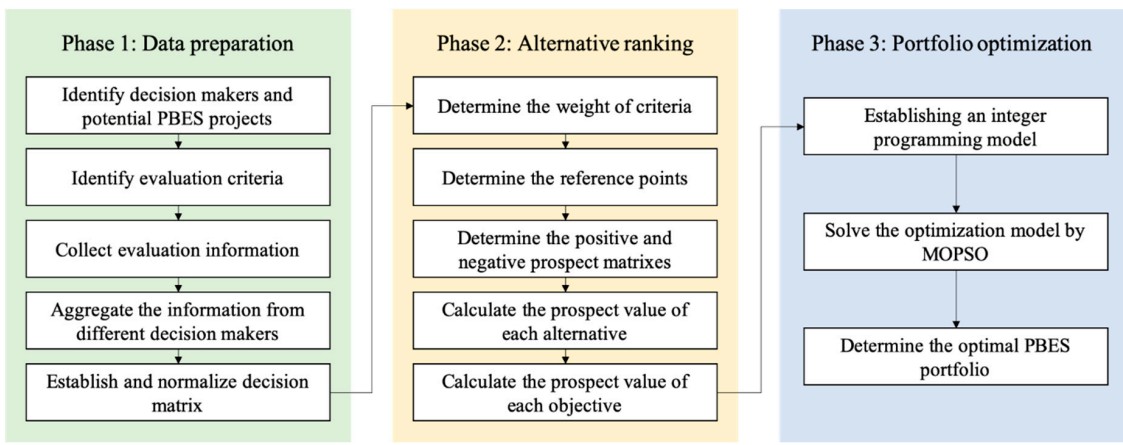

**Figure 5.** The process of the proposed decision framework.

### 4.1. Phase 1: Data Preparation

Step 1: Identifying decision-makers and candidate PBES projects.

During the decision process, some qualitative criteria which cannot be represented by crisp values are utilized to evaluate the alternatives. Therefore, it is significant to set up an expert group in which all decision-makers should have more than five years of working experience.

Meanwhile, the potential PBES projects should be investigated through multiple means such as site investigation, questionnaire, and expert consultation.

Step 2: Identifying evaluation criteria.

According to economic, social, and environmental objectives, an evaluation criteria index is established to evaluate the performance of alternatives. In this study, the criteria proposed in Section 3 will be employed in future calculations. There are two types of criteria: 1) Quantitative criteria which can be represented by crisp values, 2) qualitative criteria which contain more subjective information and can be measured by linguistic variables.

Step 3: Collecting the criteria values of each alternative.

After determining the expert panel, alternatives and criteria, let us suppose that there are $m$ alternatives $A_i(m = 1, \dots, m)$, $n$ criteria $C_j(j = 1, \dots, n)$ and $p$ decision-makers $D_k(k = 1, \dots, p)$.

The quantitative criteria values of each alternative can be collected and measured by a real number while the performance on qualitative criteria is evaluated by decision-makers using natural language phrases such as good, very good and bad, etc. Then these linguistic variables are converted into TrIFNs using a transformation rule.

Step 4: Aggregating the decision information from several decision-makers.

TrIFN-WA operator is used to integrate the information from different decision-makers and obtain the group decision values of qualitative criteria. Suppose that there are $n'$ qualitative criteria and the criteria values given by decision-makers in the form of TrIFN are $\widetilde{r}_{ij}^k = < (a_{ij}^k, b_{ij}^k, c_{ij}^k, d_{ij}^k), (a'_{ij}^k, b'_{ij}^k, c'_{ij}^k, d'_{ij}^k); \mu_{\widetilde{r}_{ij}^k}, v_{\widetilde{r}_{ij}^k} > (i = 1, \dots, m, j = 1, \dots, n', k = 1, \dots, p)$. Then the aggregation operator is defined by Equation (16).

$$
\begin{aligned}
TrINF - WA(\widetilde{r}_{ij}^1, \widetilde{r}_{ij}^2, \dots, \widetilde{r}_{ij}^p) = \sum_{k=1}^{p} \lambda_k \widetilde{r}_{ij}^k =< (\sum_{k=1}^{p} \lambda_k a_{ij}^k, \sum_{k=1}^{p} \lambda_k b_{ij}^k, \sum_{k=1}^{p} \lambda_k c_{ij}^k, \sum_{k=1}^{p} \lambda_k a_{ij}^k), \\
(\sum_{k=1}^{p} \lambda_k a'_{ij}^k, \sum_{k=1}^{p} \lambda_k b_{ij}^k, \sum_{k=1}^{p} \lambda_k c_{ij}^k, \sum_{k=1}^{p} \lambda_k d'_{ij}^k); \min \mu_{\widetilde{r}_{ij}^k}, \max v_{\widetilde{r}_{ij}^k} >
\end{aligned}
\tag{16}
$$

where $\lambda_k$ refers to the weight of the kth decision-maker. In this study, the decision-makers are equally important, which means they have the same weight.

Step 5: Establishing and normalizing the decision matrix.

In order to maintain consistency in the following calculation, the quantitative criteria values denoted by real numbers need to be transformed into TrIFNs. For example, 5 can be converted into $\langle (5,5,5,5); 1,0 \rangle$. After obtaining the aggregated decision values of quantitative criteria, the decision matrix can be established and expressed as $H = \left(\widetilde{h}_{ij}\right)_{m \times n}$ where $\widetilde{h}_{ij} = < (h_{ij}^1, h_{ij}^2, h_{ij}^3, h_{ij}^4), (h_{ij}^1{}', h_{ij}^2, h_{ij}^3, h_{ij}^4{}'); \mu_{\widetilde{h}_{ij}}, v_{\widetilde{h}_{ij}} >$.

Considering the effect of different dimensions on the final result, the decision matrix should be normalized into $X = \left(\widetilde{x}_{ij}\right)_{m \times n}$ where $\widetilde{x}_{ij} = < (x_{ij}^1, x_{ij}^2, x_{ij}^3, x_{ij}^4), (x_{ij}^1{}', x_{ij}^2, x_{ij}^3, x_{ij}^4{}'); \mu_{\widetilde{x}_{ij}}, v_{\widetilde{x}_{ij}} >$.

The set of criteria can be divided into two subsets: benefit criteria set $C_b$ and cost criteria set $C_c$. For benefit criteria, we have

$$
x_{ij}^t = \frac{h_{ij}^t - \min\limits_{i}\left\{h_{ij}^1, h_{ij}^1{}'\right\}}{\max\limits_{i}\left\{h_{ij}^4, h_{ij}^4{}'\right\} - \min\limits_{i}\left\{h_{ij}^1, h_{ij}^1{}'\right\}} \quad t = 1, 2, 3, 4. \quad c_j \in C_b
\tag{17}
$$

$$
x_{ij}^t{}' = \frac{h_{ij}^t{}' - \min\limits_{i}\left\{h_{ij}^1, h_{ij}^1{}'\right\}}{\max\limits_{i}\left\{h_{ij}^4, h_{ij}^4{}'\right\} - \min\limits_{i}\left\{h_{ij}^1, h_{ij}^1{}'\right\}} \quad t = 1, 2, 3, 4. \quad c_j \in C_b
\tag{18}
$$

For cost criteria, the normalization is defined as follows.

$$x_{ij}^t = \frac{\max\limits_i \left\{ h_{ij}^4, h_{ij}^{4\prime} \right\} - h_{ij}^{5-t}}{\max\limits_i \left\{ h_{ij}^4, h_{ij}^{4\prime} \right\} - \min\limits_i \left\{ h_{ij}^1, h_{ij}^{1\prime} \right\}} \quad t = 1, 2, 3, 4. \quad c_j \in C_c \tag{19}$$

$$x_{ij}^{t\,\prime} = \frac{\max\limits_i \left\{ h_{ij}^4, h_{ij}^{4\prime} \right\} - h_{ij}^{5-t\prime}}{\max\limits_i \left\{ h_{ij}^4, h_{ij}^{4\prime} \right\} - \min\limits_i \left\{ h_{ij}^1, h_{ij}^{1\prime} \right\}} \quad t = 1, 2, 3, 4. \quad c_j \in C_c \tag{20}$$

Here, $x_{ij}^2 = x_{ij}^{2\prime}$ and $x_{ij}^3 = x_{ij}^{3\prime}$.

## 4.2. Phase 2: Alternative Ranking and Objective Value Calculation

Step 6: Determining the weights of criteria.

In this step, the weights of criteria are determined by the entropy weight method, which is based on Shannon entropy proposed in 1948. The importance of criteria can be measured objectively by using this method. First, the normalized fuzzy values of alternatives with respect to each criterion need to be defuzzified into $G = \left\{ g_{ij} \right\}_{m \times n}$ using Equation (2). Then the entropy of each criterion is calculated according to Equation (21).

$$h_j = -K \sum_{i=1}^m t_{ij} \ln t_{ij} \tag{21}$$

where $K = 1 / \ln m$ and $t_{ij} = g_{ij} / \sum_{i=1}^m g_{ij}$.

Finally, the weights of criteria can be determined by Equation (22).

$$\omega_j = \frac{1 - h_j}{\sum_{j=1}^n (1 - h_j)} \quad j = 1, \ldots, n \tag{22}$$

Step 7: Determining the reference points.

The reference points are important in cumulative prospect theory, they represent the attitudes of decision-makers to risks. In this study, based on the concept of the TOPSIS method, the positive ideal solution (PIS) and negative ideal solution (NIS) are considered as reference points [28]. Then the PIS $S^+ = \left\{ \widetilde{b}_1^+, \widetilde{b}_2^+, \ldots, \widetilde{b}_n^+ \right\}$ and NIS $S^- = \left\{ \widetilde{b}_1^-, \widetilde{b}_2^-, \ldots, \widetilde{b}_n^- \right\}$ can be determined according to Equations (23) and (24).

$$\widetilde{b}_j^+ = < (\max_i x_{ij}^1, \max_i x_{ij}^2, \max_i x_{ij}^3, \max_i x_{ij}^4), (\max_i x_{ij}^{1\prime}, \max_i x_{ij}^2, \max_i x_{ij}^3, \max_i x_{ij}^{4\prime}); \max_i \mu_{\widetilde{x}_{ij}}, \min_i v_{\widetilde{x}_{ij}} > \tag{23}$$

$$\widetilde{b}_j^- = < (\min_i x_{ij}^1, \min_i x_{ij}^2, \min_i x_{ij}^3, \min_i x_{ij}^4), (\min_i x_{ij}^{1\prime}, \min_i x_{ij}^2, \min_i x_{ij}^3, \min_i x_{ij}^{4\prime}); \min_i \mu_{\widetilde{x}_{ij}}, \max_i v_{\widetilde{x}_{ij}} > \tag{24}$$

Step 8: Determining the positive and negative prospect matrixes.

According to the reference points, the value function can be expressed as follows.

$$v(\widetilde{x}_{ij}) = \begin{cases} \left[ d(\widetilde{x}_{ij}, \widetilde{b}_j^-) \right]^\alpha & \text{if } S^- \text{ is the reference point} \\ -\theta \left[ d(\widetilde{x}_{ij}, \widetilde{b}_j^+) \right]^\beta & \text{if } S^+ \text{ is the reference point} \end{cases} \tag{25}$$

When $S^-$ is the reference point, decision-makers are confronted with gains, and the positive prospect matrix can be obtained according to Equation (25). On the contrary, the reference point $S^+$ means that the decision-makers are inclined to risk aversion and, in this case, the negative prospect matrix can be calculated.

Step 9: Calculating the prospect value of each alternative.

Based on the criteria weights and prospect matrixes, the prospect value of each alternative can be obtained according to Equation (26).

$$V_i = \sum_{j=1}^{n} \pi^+(\omega_j) v^+(\widetilde{x}_{ij}) + \sum_{j=1}^{n} \pi^-(\omega_j) v^-(\widetilde{x}_{ij}) \tag{26}$$

From an overall perspective, the alternative with higher prospect value performs better on the general objective. However, in the proposed portfolio optimization problem, various objectives are considered, and multiple alternatives will be selected. Therefore, the prospect value of each alternative with respect to the economic, social, and environmental objective needs to be computed according to Equation (27).

$$V_i^t = \sum_{j=1}^{n_t} \pi^+(\omega_j) v^+(\widetilde{x}_{ij}) + \sum_{j=1}^{n_t} \pi^-(\omega_j) v^-(\widetilde{x}_{ij}) \quad t = 1,2,3 \tag{27}$$

where $V_i^1$, $V_i^2$ and $V_i^3$ refer to the prospect value of the ith alternative with respect to economic, social and environmental objective respectively, and $n_1$, $n_2$, and $n_3$ are the economic factor set, social factor set and environmental factor set, respectively.

### 4.3. Phase 3: Portfolio Optimization

Step 10: Establishing an integer programming model.

In order to find out the best investment portfolio for PBES, a multi-objective 0-1 integer programming model is established as follows.

$$f_t = \max \left\{ \sum_{i=1}^{m} x_i V_i^t \right\} \quad t = 1,2,3$$

$$s.t. \begin{cases} \sum_{i=1}^{m} x_i C_i \leq C \\ x_i = \begin{cases} 1 & \textit{if selected} \\ 0 & \textit{if not selected} \end{cases} \end{cases} \tag{28}$$

where $f_t$ is the tth objective (t = 1,2,3), $C_i$ is the total cost of the ith alternative and $C$ is the cost constraint.

Step 11: Solve the optimization model by MOPSO and determine the best solution.

The above optimization model can be solved by using the MOPSO method, and then the optimal portfolio for PBES can be obtained. In this study, the parameters of MOPSO are given in Table 1 [29].

**Table 1.** The parameters of MOPSO.

| Parameter | Value |
|---|---|
| Population | 100 |
| Number of generations | 100 |
| $c_1$ | 2.5 |
| $c_2$ | 2.5 |
| $\omega_{max}$ | 0.8 |
| $\omega_{min}$ | 0.4 |
| Mutation rate | 0.3 |

## 5. Case Study

Based on the proposed criteria system and decision framework, a case in China will be studied and discussed in this section.

### 5.1. Problem Statement

An energy investment company plans to build several PBESs in some provinces of South China. After field investigation, the management team and the expert group identified 17 potential projects, and 10 of them proved to be economically feasible. Then, the best portfolio will be determined among these feasible alternatives. The geographical graph of 10 PBES projects is shown in Figure 6.

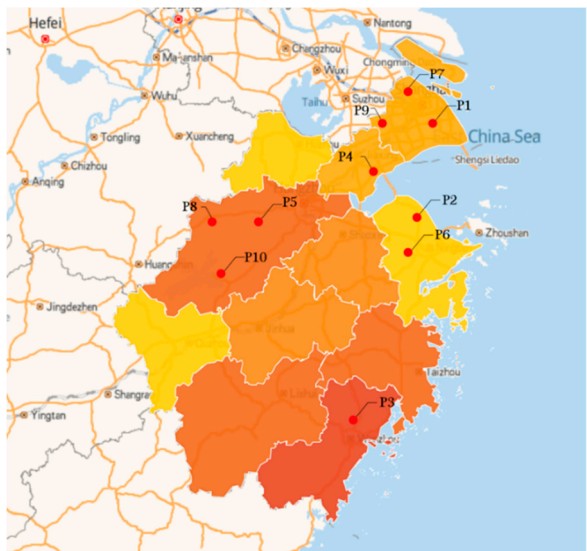

**Figure 6.** The geographical distribution of 10 feasible PBES projects.

In this case, three decision-makers are invited to evaluate the performance of PBES alternatives based on their professional knowledge and working experience. The profiles of decision-makers are presented in Table 2.

**Table 2.** The profiles of decision-makers.

| No. | Gender | Occupation | Age | Years of Working |
|-----|--------|------------|-----|------------------|
| D1 | Male | University professor in the field of energy management | 48 | 18 |
| D2 | Female | Project manager in an EV company | 40 | 17 |
| D3 | Male | Department head in a state-owned grid enterprise | 48 | 25 |

### 5.2. Optimization Result and Result Analysis

In this study, 14 decision-making criteria are identified in Section 2, which can be classified into benefit and cost type. C11, C12, C14, C33, and C34 are cost type, while the others are benefit. In addition, C11, C12, C13, C14, and C32 are quantitative criteria that are collected from field data or predicted by experts while the other criteria are qualitative which need to be measured by decision-makers according to the linguistic terms in Table 3. With respect to the qualitative criteria, the linguistic variables should be converted into TrIFNs based on the transformation rule proposed in Table 3.

After collecting the evaluation data, the performance of 10 feasible PBES projects with respect to each criterion is obtained and presented in Table 4.

Then, transform the given linguistic variables into TrIFNs and use TrIFN-WA operator to integrate the comments of three decision-makers. According to Equations (10)–(13), the normalized decision matrix is obtained, shown in Table A1 in Appendix A.

**Table 3.** The transformation rule between linguistic variables and TrIFNs.

| Linguistic Variables | TrIFN |
|---|---|
| Very low (VL) | <(0.05, 0.10, 0.15, 0.17), (0.00, 0.10, 0.15, 0.20); 0.10, 0.90]> |
| Low (L) | <(0.15, 0.20, 0.25, 0.30), (0.10, 0.20, 0.25, 0.35); 0.20, 0.75]> |
| Medium low (ML) | <(0.30, 0.35, 0.40, 0.45), (0.25, 0.35, 0.40, 0.50); 0.35, 0.60]> |
| Medium (M) | <(0.45, 0.50, 0.55, 0.57), (0.40, 0.50, 0.55, 0.60); 0.50, 0.50]> |
| Medium high (MH) | <(0.59, 0.60, 0.70, 0.74), (0.58, 0.60, 0.70, 0.75); 0.65, 0.25]> |
| High (H) | <(0.77, 0.80, 0.85, 0.87), (0.75, 0.80, 0.85, 0.90); 0.80, 0.15]> |
| Very high (VH) | <(0.87, 0.90, 0.95, 0.97), (0.85, 0.90, 0.95, 1.00); 0.95, 0.05]> |

**Table 4.** The performance of alternatives with respect to each criterion.

|  | Unit |  | P1 | P2 | P3 | P4 | P5 | P6 | P7 | P8 | P9 | P10 |
|---|---|---|---|---|---|---|---|---|---|---|---|---|
| C11 | $10^5$USD |  | 7.05 | 9.30 | 5.54 | 10.42 | 7.30 | 14.24 | 9.21 | 8.04 | 7.26 | 9.40 |
| C12 | $10^4$USD |  | 9.30 | 11.42 | 6.50 | 8.72 | 8.40 | 13.43 | 10.20 | 7.20 | 9.35 | 10.30 |
| C13 | $10^5$USD |  | 5.20 | 4.36 | 2.30 | 4.70 | 4.20 | 7.25 | 4.60 | 3.85 | 3.70 | 4.93 |
| C14 | Year |  | 5.82 | 8.68 | 9.47 | 6.86 | 6.74 | 6.60 | 7.55 | 6.76 | 8.28 | 7.13 |
|  | - | D1 | M | M | H | H | H | MH | H | H | MH | VH |
| C21 | - | D2 | MH | M | MH | VH | H | H | H | MH | M | MH |
|  | - | D3 | M | ML | H | H | H | MH | MH | H | M | VH |
|  | - | D1 | ML | MH | MH | MH | M | MH | H | M | H | VH |
| C22 | - | D2 | M | H | MH | M | H | M | H | VH | H | MH |
|  | - | D3 | ML | M | MH | H | H | M | H | VH | H | VH |
|  | - | D1 | MH | M | M | MH | VH | VH | MH | MH | MH | MH |
| C23 | - | D2 | M | M | VH | VH | H | MH | VH | VH | H | M |
|  | - | D3 | H | MH | M | MH | VH | M | H | H | VH | MH |
|  | - | D1 | VH | MH | H | VH | M | MH | H | VH | MH | M |
| C24 | - | D2 | H | M | MH | MH | M | MH | MH | M | MH | MH |
|  | - | D3 | MH | VH | H | MH | ML | MH | MH | H | MH | M |
|  | - | D1 | ML | MH | MH | H | M | M | M | H | M | M |
| C25 | - | D2 | M | H | H | MH | ML | MH | ML | MH | MH | VH |
|  | - | D3 | M | H | MH | H | L | H | M | H | VH | MH |
|  | - | D1 | M | H | ML | M | M | ML | ML | MH | H | MH |
| C26 | - | D2 | ML | MH | L | H | ML | M | M | VH | ML | M |
|  | - | D3 | M | MH | M | M | M | ML | MH | VH | L | VH |
|  | - | D1 | MH | H | H | ML | H | MH | H | ML | H | M |
| C31 | - | D2 | H | MH | H | M | VH | M | M | M | M | MH |
|  | - | D3 | VH | MH | MH | ML | VH | MH | MH | L | M | MH |
| C32 | t |  | 350 | 382 | 264 | 425 | 367 | 720 | 415 | 373 | 365 | 403 |
|  | - | D1 | L | M | M | MH | L | ML | L | L | M | ML |
| C33 | - | D2 | VL | ML | M | L | M | ML | ML | M | L | VL |
|  | - | D3 | L | ML | ML | ML | VL | L | ML | M | ML | VL |
|  | - | D1 | VL | VL | ML | ML | M | L | ML | VL | L | M |
| C34 | - | D2 | M | L | ML | M | ML | L | L | L | ML | H |
|  | - | D3 | ML | VL | VL | L | ML | ML | L | L | VL | M |

According to Equations (14) and (15), the weights of 14 criteria are calculated, shown in Table 5. It can be seen from Table 5 that a reduction in GHG emissions (C32) has the highest importance. Meanwhile, the weights of the three objectives are also obtained with the values of 0.30, 0.42, and 0.28 respectively, which means the social objective is more important than the other objectives.

**Table 5.** The criteria weights.

| C11 | C12 | C13 | C14 | C21 | C22 | C23 | C24 | C25 | C26 | C31 | C32 | C33 | C34 |
|---|---|---|---|---|---|---|---|---|---|---|---|---|---|
| 0.06 | 0.08 | 0.08 | 0.08 | 0.06 | 0.06 | 0.07 | 0.04 | 0.08 | 0.12 | 0.09 | 0.13 | 0.02 | 0.03 |

According to Equations (16) and (17), PIS and NIS can be obtained. Then, the distance between alternatives and PIS or NIS is calculated. Subsequently, the positive and negative prospect value matrixes are determined based on Equation (18), presented in Tables 6 and 7.

**Table 6.** Positive prospect value matrix.

|     | P1 | P2 | P3 | P4 | P5 | P6 | P7 | P8 | P9 | P10 |
|-----|------|------|------|------|------|------|------|------|------|------|
| C11 | 0.85 | 0.61 | 1.00 | 0.48 | 0.82 | 0.00 | 0.62 | 0.74 | 0.82 | 0.60 |
| C12 | 0.63 | 0.34 | 1.00 | 0.71 | 0.75 | 0.00 | 0.51 | 0.91 | 0.63 | 0.50 |
| C13 | 0.62 | 0.46 | 0.00 | 0.53 | 0.43 | 1.00 | 0.51 | 0.36 | 0.33 | 0.57 |
| C14 | 1.00 | 0.26 | 0.00 | 0.75 | 0.78 | 0.81 | 0.57 | 0.77 | 0.37 | 0.68 |
| C21 | 0.11 | 0.00 | 0.45 | 0.65 | 0.61 | 0.38 | 0.45 | 0.45 | 0.11 | 0.52 |
| C22 | 0.00 | 0.19 | 0.38 | 0.27 | 0.31 | 0.18 | 0.66 | 0.37 | 0.66 | 0.57 |
| C23 | 0.13 | 0.00 | 0.12 | 0.34 | 0.64 | 0.17 | 0.42 | 0.42 | 0.42 | 0.07 |
| C24 | 0.55 | 0.42 | 0.51 | 0.48 | 0.00 | 0.37 | 0.44 | 0.34 | 0.37 | 0.15 |
| C25 | 0.12 | 0.32 | 0.54 | 0.60 | 0.00 | 0.35 | 0.12 | 0.60 | 0.38 | 0.38 |
| C26 | 0.11 | 0.26 | 0.00 | 0.28 | 0.11 | 0.08 | 0.14 | 0.61 | 0.05 | 0.34 |
| C31 | 0.54 | 0.46 | 0.51 | 0.07 | 0.73 | 0.26 | 0.30 | 0.00 | 0.26 | 0.26 |
| C32 | 0.23 | 0.30 | 0.00 | 0.40 | 0.27 | 1.00 | 0.38 | 0.28 | 0.27 | 0.35 |
| C33 | 0.08 | 0.02 | 0.08 | 0.07 | 0.05 | 0.12 | 0.12 | 0.07 | 0.09 | 0.08 |
| C34 | 0.06 | 0.10 | 0.07 | 0.13 | 0.18 | 0.17 | 0.17 | 0.09 | 0.08 | 0.07 |

**Table 7.** Negative prospect value matrix.

|     | P1 | P2 | P3 | P4 | P5 | P6 | P7 | P8 | P9 | P10 |
|-----|-------|-------|-------|-------|-------|-------|-------|-------|-------|-------|
| C11 | −0.55 | −1.22 | 0 | −1.53 | −0.62 | −2.55 | −1.19 | −0.85 | −0.61 | −1.25 |
| C12 | −1.15 | −1.89 | 0 | −0.94 | −0.82 | −2.55 | −1.47 | −0.34 | −1.17 | −1.5 |
| C13 | −1.17 | −1.59 | −2.55 | −1.42 | −1.67 | 0 | −1.47 | −1.83 | −1.9 | −1.31 |
| C14 | 0 | −2.06 | −2.55 | −0.84 | −0.76 | −0.65 | −1.32 | −0.77 | −1.8 | −1.03 |
| C21 | −1.47 | −1.66 | −0.66 | 0 | −0.17 | −0.83 | −0.66 | −0.66 | −1.47 | −0.44 |
| C22 | −1.72 | −1.37 | −0.9 | −1.18 | −1.07 | −1.39 | −0.06 | −0.94 | −0.06 | −0.36 |
| C23 | −1.38 | −1.63 | −1.42 | −0.9 | 0 | −1.3 | −0.7 | −0.7 | −0.7 | −1.51 |
| C24 | 0 | −0.44 | −0.16 | −0.25 | −1.41 | −0.59 | −0.38 | −0.66 | −0.59 | −1.12 |
| C25 | −1.31 | −0.83 | −0.23 | 0 | −1.53 | −0.77 | −1.31 | 0 | −0.7 | −0.7 |
| C26 | −1.36 | −1.02 | −1.56 | −0.98 | −1.36 | −1.43 | −1.3 | 0 | −1.48 | −0.82 |
| C31 | −0.61 | −0.84 | −0.7 | −1.74 | 0 | −1.35 | −1.26 | −1.86 | −1.34 | −1.35 |
| C32 | −2.12 | −1.96 | −2.55 | −1.74 | −2.04 | 0 | −1.79 | −2.01 | −2.05 | −1.85 |
| C33 | −0.64 | −0.77 | −0.64 | −0.67 | −0.69 | −0.55 | −0.55 | −0.66 | −0.61 | −0.65 |
| C34 | −1.03 | −0.96 | −1.01 | −0.88 | −0.76 | −0.8 | −0.8 | −0.97 | −0.99 | −1.01 |

Based on Equations (19) and (20), the prospect values of each objective and the final prospect values are calculated respectively. The result is shown in Table 8.

**Table 8.** The prospect values of alternatives.

|       | P1 | P2 | P3 | P4 | P5 | P6 | P7 | P8 | P9 | P10 |
|-------|-------|-------|-------|-------|-------|-------|-------|-------|-------|-------|
| O1 | 0.09 | −0.71 | −0.46 | −0.27 | −0.12 | −0.48 | −0.43 | −0.1 | −0.46 | −0.35 |
| O2 | −0.89 | −0.77 | −0.45 | −0.09 | −0.52 | −0.67 | −0.33 | 0.09 | −0.43 | −0.35 |
| O3 | −0.49 | −0.5 | −0.64 | −0.62 | −0.3 | −0.04 | −0.5 | −0.75 | −0.62 | −0.58 |
| Total | −1.29 | −1.98 | −1.55 | −0.98 | −0.94 | −1.19 | −1.26 | −0.76 | −1.51 | −1.28 |

In terms of economic objective, P1 has the highest prospect value, which means P1 has the best economic performance. Moreover, P8 ranks first on the social objective while P6 performs best on the environmental objective. For the overall performance, P8 has the highest prospect value of −0.76. The rank of 10 alternatives is P8>P5>P4>P6>P7>P10>P1>P9>P3>P2.

In this case, the optimal portfolio will be selected on a limited budget of less than three million dollars. Since most objective prospect values are negative, they should be normalized before optimization. Then, solve the proposed optimization model in Equation (21) using MOPSO. Finally, 13 non-dominated solutions are obtained, shown in Figure 7 and Table 9.

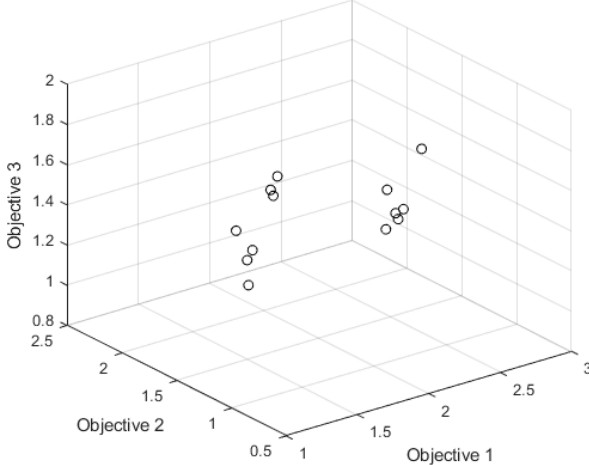

**Figure 7.** The optimization results.

**Table 9.** The final non−dominated solutions.

| No. | P1 | P2 | P3 | P4 | P5 | P6 | P7 | P8 | P9 | P10 | O1 | O2 | O3 |
|-----|----|----|----|----|----|----|----|----|----|-----|------|------|------|
| 1 | 0 | 0 | 1 | 0 | 1 | 0 | 0 | 1 | 1 | 0 | 2.11 | 2.31 | 0.94 |
| 2 | 1 | 1 | 1 | 0 | 1 | 0 | 0 | 0 | 0 | 0 | 2.05 | 0.96 | 1.48 |
| 3 | 0 | 0 | 0 | 0 | 1 | 1 | 0 | 1 | 0 | 0 | 1.79 | 1.60 | 1.62 |
| 4 | 1 | 0 | 1 | 0 | 1 | 0 | 0 | 0 | 0 | 1 | 2.50 | 1.38 | 1.37 |
| 5 | 0 | 0 | 1 | 0 | 1 | 0 | 1 | 0 | 1 | 0 | 1.71 | 1.88 | 1.29 |
| 6 | 0 | 0 | 0 | 0 | 1 | 1 | 0 | 0 | 1 | 0 | 1.34 | 1.08 | 1.79 |
| 7 | 0 | 0 | 0 | 0 | 1 | 0 | 1 | 1 | 0 | 0 | 1.85 | 1.95 | 0.97 |
| 8 | 0 | 0 | 1 | 0 | 1 | 0 | 0 | 0 | 1 | 1 | 1.85 | 1.86 | 1.18 |
| 9 | 1 | 0 | 0 | 0 | 1 | 1 | 0 | 0 | 0 | 0 | 2.03 | 0.60 | 1.98 |
| 10 | 0 | 0 | 1 | 0 | 1 | 1 | 0 | 0 | 0 | 0 | 1.34 | 1.05 | 1.77 |
| 11 | 1 | 0 | 1 | 0 | 1 | 0 | 0 | 1 | 0 | 0 | 2.82 | 1.83 | 1.13 |
| 12 | 1 | 0 | 1 | 0 | 1 | 0 | 1 | 0 | 0 | 0 | 2.40 | 1.41 | 1.48 |
| 13 | 1 | 0 | 0 | 0 | 1 | 0 | 0 | 1 | 1 | 0 | 2.81 | 1.85 | 1.16 |

Theoretically, the optimization result is a set of Pareto solutions, which means each one of them cannot dominate another. In real cases, if the investors need fewer solutions to make decisions, they can select the desirable solutions using methods that meet their requirements. For example, based on the weight of three objectives, the weighted objective value of each solution can be calculated, and the best one can be selected. The methods may vary with the requirement of investors, so this part will not be discussed in this study.

*5.3. Discussion*

Scenario analysis is conducted to investigate the optimization result in different scenarios. The proposed case is considered as the base scenario. Then, considering different objective combinations, six scenarios are considered and described as follows.

Base scenario: Considering economic, social, and environmental objectives.

Scenario 1: Considering economic and social objectives (see Figure 8a).

Scenario 2: Considering economic and environmental objectives (see Figure 8b).

Scenario 3: Considering social and environmental objectives (see Figure 8c).

Scenario 4: Only considering economic objective (see Figure 9).
Scenario 5: Only considering social objective (see Figure 9).
Scenario 6: Only considering environmental objective (see Figure 9).

In scenarios 1–3, two objectives are taken into consideration. The optimization results of these three scenarios are shown in Figure 8. As shown in Figure 8, scenario 1 has one optimal solution while scenario 2 and 3 have 5 and 6 Pareto solutions respectively.

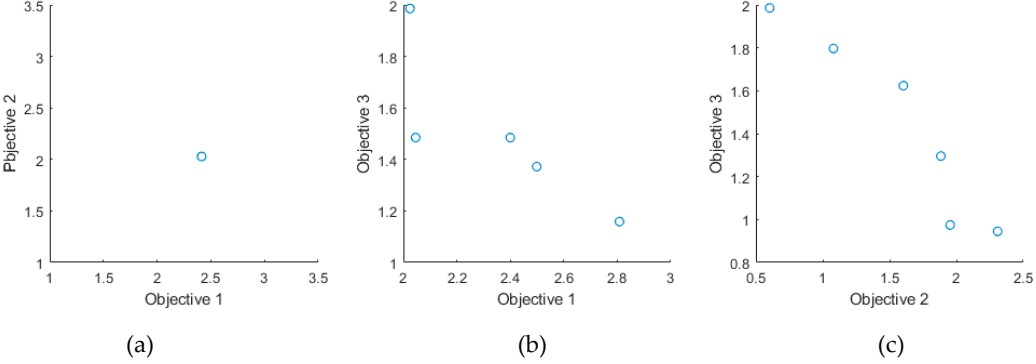

(a)                      (b)                      (c)

**Figure 8.** Results of scenarios 1–3.

Scenarios 4–6 only consider one objective and the results are illustrated in Figure 9. The result shows each scenario has one best solution.

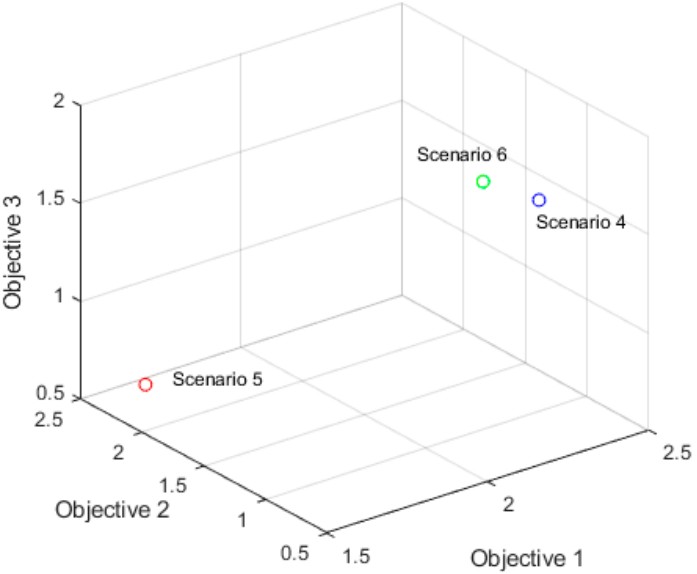

**Figure 9.** Results of scenarios 4–6.

The optimal portfolios in six scenarios are presented in Table 10.

These six scenarios obtain fewer Pareto solutions than the base scenario. It also can be seen from Table 10 that the solutions of scenarios 2, 3, 4, and 6 can be found in the result of the base scenario. The results of scenario 1 and 5 have the best performance on one or two objectives but still can be dominated by Pareto solutions in the base scenario. Therefore, the sustainable scenario considering economic, social, and environmental objectives can provide more satisfactory solutions for decision-makers, which is more reasonable than other scenarios.

**Table 10.** The optimal portfolios of each scenario.

|  | P1 | P2 | P3 | P4 | P5 | P6 | P7 | P8 | P9 | P10 |
|---|---|---|---|---|---|---|---|---|---|---|
| Scenario 1 | 1 | 0 | 1 | 0 | 0 | 0 | 1 | 1 | 0 | 0 |
|  | 1 | 0 | 1 | 0 | 1 | 0 | 0 | 0 | 0 | 1 |
|  | 1 | 0 | 0 | 0 | 1 | 0 | 0 | 1 | 1 | 0 |
| Scenario 2 | 1 | 0 | 1 | 0 | 1 | 0 | 1 | 0 | 0 | 0 |
|  | 1 | 0 | 0 | 0 | 1 | 1 | 0 | 0 | 0 | 0 |
|  | 1 | 1 | 1 | 0 | 1 | 0 | 0 | 0 | 0 | 0 |
|  | 0 | 0 | 1 | 0 | 1 | 0 | 0 | 1 | 1 | 0 |
|  | 0 | 0 | 0 | 0 | 1 | 1 | 0 | 1 | 0 | 0 |
| Scenario 3 | 0 | 0 | 0 | 0 | 1 | 1 | 0 | 0 | 1 | 0 |
|  | 0 | 0 | 1 | 0 | 1 | 0 | 1 | 0 | 1 | 0 |
|  | 1 | 0 | 0 | 0 | 1 | 1 | 0 | 0 | 0 | 0 |
|  | 0 | 0 | 0 | 0 | 1 | 0 | 1 | 1 | 0 | 0 |
| Scenario 4 | 1 | 0 | 1 | 0 | 1 | 0 | 0 | 0 | 0 | 1 |
| Scenario 5 | 0 | 0 | 0 | 1 | 0 | 0 | 1 | 1 | 0 | 0 |
| Scenario 6 | 1 | 0 | 0 | 0 | 1 | 1 | 0 | 0 | 0 | 0 |

## 6. Conclusions

PBES is a PV−assisted EV charging station that makes better use of PV resources and reduces the stress of charging demand on the utility grid. Consequently, an increasing number of enterprises intend to invest in PBES projects. This paper proposed a PBES portfolio optimization framework from a sustainable perspective to provide investment decisions for investors. The main conclusions of this study are summarized as follows.

(1) From a sustainable perspective, this paper identified 14 decision-making criteria considering economy, society, and environment. Among these criteria, C11, C12, C14, C33, and C34 are cost type while the others are benefit. In addition, C11, C12, C13, C14, and C32 are quantitative criteria, while others are qualitative criteria.

(2) A hybrid PBES portfolio optimization framework under trapezoidal intuitionistic fuzzy environment was proposed. Firstly, the evaluation of each alternative with respect to each criterion is converted to TrIFNs. Then, alternatives are ranked based on cumulative prospect theory, and the objective values are obtained. Finally, a multi-objective optimization model is established and solved by MOPSO.

(3) In the case study, 10 feasible PBES projects in South China were selected as alternatives. The decision-making result proved that C32 was the most important criterion and P8 had the best overall performance. After solving the optimization model, 13 non-dominated Pareto solutions were obtained.

(4) A scenario analysis was conducted to verify the superiority of the proposed model. The optimization results of six scenarios were obtained and compared with the result of the base scenario. The result shows that the Pareto solutions in the base scenario include the results of scenario 2, 3, 4, and 6, which means the proposed model can obtain more satisfactory solutions in a sustainable scenario.

In the future study, this portfolio optimization model can be applied to other project portfolio selection problems. Moreover, how to deal with vague information in the portfolio optimization still needs to be studied deeply.

**Author Contributions:** Methodology, case study and writing, Q.D.; conceptualization and data collection, J.L. All authors have read and agreed to the published version of the manuscript.

**Funding:** This research was funded by National Natural Science Foundation of China (Grant number: 71771085), Scientific Research Project of Universities in Inner Mongolia (Grant number: NJZC17415) and Research Project of Ordos Institute of Technology (Grant number: KYYB2017013).

**Acknowledgments:** The authors would like to thank the editor of this journal and the reviewers for their detailed and helpful comments.

**Conflicts of Interest:** The authors declare no conflict of interest.

# Appendix A

**Table A1.** The normalized decision matrix.

|  | P1 | 2 | P3 | P4 | P5 |
|---|---|---|---|---|---|
| C11 | <(0.83,0.83,0.83,0.83),(0.83,0.83,0.83,0.83);1,0> | <(0.57,0.57,0.57,0.57),(0.57,0.57,0.57,0.57);1,0> | <(1,1,1,1),(1,1,1,1);1,0> | <(0.44,0.44,0.44,0.44),(0.44,0.44,0.44,0.44);1,0> | <(0.8,0.8,0.8,0.8),(0.8,0.8,0.8,0.8);1,0> |
| C12 | <(0.6,0.6,0.6,0.6),(0.6,0.6,0.6,0.6);1,0> | <(0.29,0.29,0.29,0.29),(0.29,0.29,0.29,0.29);1,0> | <(1,1,1,1),(1,1,1,1);1,0> | <(0.68,0.68,0.68,0.68),(0.68,0.68,0.68,0.68);1,0> | <(0.73,0.73,0.73,0.73),(0.73,0.73,0.73,0.73);1,0> |
| C13 | <(0.59,0.59,0.59,0.59),(0.59,0.59,0.59,0.59);1,0> | <(0.42,0.42,0.42,0.42),(0.42,0.42,0.42,0.42);1,0> | <(0,0,0,0),(0,0,0,0);1,0> | <(0.48,0.48,0.48,0.48),(0.48,0.48,0.48,0.48);1,0> | <(0.38,0.38,0.38,0.38),(0.38,0.38,0.38,0.38);1,0> |
| C14 | <(1,1,1,1),(1,1,1,1);1,0> | <(0.22,0.22,0.22,0.22),(0.22,0.22,0.22,0.22);1,0> | <(0,0,0,0),(0,0,0,0);1,0> | <(0.72,0.72,0.72,0.72),(0.72,0.72,0.72,0.72);1,0> | <(0.75,0.75,0.75,0.75),(0.75,0.75,0.75,0.75);1,0> |
| C21 | <(0.26,0.32,0.43,0.48),(0.19,0.32,0.43,0.52);0.5,0.5> | <(0.09,0.18,0.26,0.31),(0,0.18,0.26,0.38);0.5,0.5> | <(0.62,0.66,0.77,0.82),(0.59,0.66,0.77,0.86);0.5,0.5> | <(0.78,0.83,0.91,0.95),(0.74,0.83,0.91,1);0.8,0.15> | <(0.72,0.77,0.86,0.89),(0.69,0.77,0.86,0.94);0.8,0.15> |
| C22 | <(0.09,0.17,0.25,0.32),(0,0.17,0.25,0.39);0.35,0.6> | <(0.5,0.55,0.65,0.7),(0.45,0.55,0.65,0.73);0.35,0.6> | <(0.48,0.49,0.65,0.72),(0.46,0.49,0.65,0.73);0.35,0.6> | <(0.5,0.55,0.65,0.7),(0.45,0.55,0.65,0.73);0.5,0.5> | <(0.59,0.65,0.73,0.76),(0.55,0.65,0.73,0.81);0.5,0.5> |
| C23 | <(0.28,0.34,0.47,0.53),(0.23,0.34,0.47,0.57);0.5,0.5> | <(0.07,0.14,0.28,0.33),(0,0.14,0.28,0.38);0.5,0.5> | <(0.26,0.34,0.44,0.48),(0.18,0.34,0.44,0.54);0.5,0.5> | <(0.44,0.47,0.64,0.7),(0.41,0.47,0.64,0.74);0.65,0.25> | <(0.74,0.8,0.9,0.94),(0.7,0.8,0.9,1);0.8,0.15> |
| C24 | <(0.74,0.78,0.91,0.96),(0.71,0.78,0.91,1);0.65,0.25> | <(0.54,0.6,0.72,0.77),(0.49,0.6,0.72,0.81);0.65,0.25> | <(0.68,0.72,0.84,0.89),(0.65,0.72,0.84,0.94);0.65,0.25> | <(0.63,0.66,0.81,0.88),(0.6,0.66,0.81,0.91);0.65,0.25> | <(0.1,0.19,0.29,0.34),(0,0.19,0.29,0.41);0.35,0.6> |
| C25 | <(0.25,0.34,0.42,0.47),(0.17,0.34,0.42,0.53);0.35,0.6> | <(0.77,0.81,0.92,0.96),(0.74,0.81,0.92,1);0.35,0.6> | <(0.67,0.7,0.83,0.89),(0.65,0.7,0.83,0.92);0.35,0.6> | <(0.77,0.81,0.92,0.96),(0.74,0.81,0.92,1);0.65,0.25> | <(0.09,0.17,0.25,0.32),(0,0.17,0.25,0.39);0.2,0.75> |
| C26 | <(0.23,0.3,0.38,0.42),(0.15,0.3,0.38,0.48);0.35,0.6> | <(0.6,0.63,0.75,0.8),(0.58,0.63,0.75,0.83);0.35,0.6> | <(0.08,0.15,0.23,0.29),(0,0.15,0.23,0.35);0.35,0.6> | <(0.46,0.53,0.6,0.63),(0.4,0.53,0.6,0.68);0.5,0.5> | <(0.23,0.3,0.38,0.42),(0.15,0.3,0.38,0.48);0.35,0.6> |
| C31 | <(0.69,0.72,0.81,0.85),(0.67,0.72,0.81,0.88);0.65,0.25> | <(0.56,0.58,0.7,0.75),(0.54,0.58,0.7,0.77);0.65,0.25> | <(0.64,0.68,0.77,0.81),(0.62,0.68,0.77,0.84);0.65,0.25> | <(0.14,0.21,0.28,0.34),(0.06,0.21,0.28,0.4);0.35,0.6> | <(0.82,0.86,0.93,0.96),(0.79,0.86,0.93,1);0.8,0.15> |
| C32 | <(0.19,0.19,0.19,0.19),(0.19,0.19,0.19,0.19);1,0> | <(0.26,0.26,0.26,0.26),(0.26,0.26,0.26,0.26);1,0> | <(0,0,0,0),(0,0,0,0);1,0> | <(0.35,0.35,0.35,0.35),(0.35,0.35,0.35,0.35);1,0> | <(0.23,0.23,0.23,0.23),(0.23,0.23,0.23,0.23);1,0> |
| C33 | <(0.62,0.7,0.8,0.9),(0.53,0.7,0.8,1);0.1,0.9> | <(0.15,0.23,0.33,0.43),(0.07,0.23,0.33,0.55);0.1,0.9> | <(0.07,0.13,0.23,0.33),(0,0.13,0.23,0.44);0.1,0.9> | <(0.14,0.23,0.37,0.44),(0.07,0.23,0.37,0.52);0.2,0.75> | <(0.44,0.5,0.6,0.7),(0.37,0.5,0.6,0.8);0.1,0.9> |
| C34 | <(0.46,0.5,0.58,0.65),(0.4,0.5,0.58,0.73);0.1,0.9> | <(0.73,0.78,0.85,0.93),(0.68,0.78,0.85,1);0.1,0.9> | <(0.52,0.58,0.65,0.73),(0.45,0.58,0.65,0.81);0.1,0.9> | <(0.39,0.45,0.53,0.6),(0.33,0.45,0.53,0.68);0.2,0.75> | <(0.32,0.38,0.45,0.53),(0.25,0.38,0.45,0.61);0.35,0.6> |

|  | P6 | P7 | P8 | P9 | P10 |
|---|---|---|---|---|---|
| C11 | <(0,0,0,0),(0,0,0,0);1,0> | <(0.58,0.58,0.58,0.58),(0.58,0.58,0.58,0.58);1,0> | <(0.71,0.71,0.71,0.71),(0.71,0.71,0.71,0.71);1,0> | <(0.8,0.8,0.8,0.8),(0.8,0.8,0.8,0.8);1,0> | <(0.56,0.56,0.56,0.56),(0.56,0.56,0.56,0.56);1,0> |
| C12 | <(0,0,0,0),(0,0,0,0);1,0> | <(0.47,0.47,0.47,0.47),(0.47,0.47,0.47,0.47);1,0> | <(0.9,0.9,0.9,0.9),(0.9,0.9,0.9,0.9);1,0> | <(0.59,0.59,0.59,0.59),(0.59,0.59,0.59,0.59);1,0> | <(0.45,0.45,0.45,0.45),(0.45,0.45,0.45,0.45);1,0> |
| C13 | <(1,1,1,1),(1,1,1,1);1,0> | <(0.46,0.46,0.46,0.46),(0.46,0.46,0.46,0.46);1,0> | <(0.31,0.31,0.31,0.31),(0.31,0.31,0.31,0.31);1,0> | <(0.28,0.28,0.28,0.28),(0.28,0.28,0.28,0.28);1,0> | <(0.53,0.53,0.53,0.53),(0.53,0.53,0.53,0.53);1,0> |
| C14 | <(0.79,0.79,0.79,0.79),(0.79,0.79,0.79,0.79);1,0> | <(0.53,0.53,0.53,0.53),(0.53,0.53,0.53,0.53);1,0> | <(0.74,0.74,0.74,0.74),(0.74,0.74,0.74,0.74);1,0> | <(0.33,0.33,0.33,0.33),(0.33,0.33,0.33,0.33);1,0> | <(0.64,0.64,0.64,0.64),(0.64,0.64,0.64,0.64);1,0> |
| C21 | <(0.52,0.55,0.69,0.74),(0.49,0.55,0.69,0.77);0.65,0.25> | <(0.62,0.66,0.77,0.82),(0.59,0.66,0.77,0.86);0.65,0.25> | <(0.62,0.66,0.77,0.82),(0.59,0.66,0.77,0.86);0.65,0.25> | <(0.26,0.32,0.43,0.48),(0.19,0.32,0.43,0.52);0.5,0.5> | <(0.73,0.77,0.89,0.93),(0.7,0.77,0.89,0.97);0.65,0.25> |
| C22 | <(0.33,0.39,0.49,0.53),(0.27,0.39,0.49,0.57);0.5,0.5> | <(0.76,0.81,0.89,0.93),(0.73,0.81,0.89,0.97);0.8,0.15> | <(0.7,0.76,0.84,0.87),(0.65,0.76,0.84,0.92);0.5,0.5> | <(0.76,0.81,0.89,0.93),(0.73,0.81,0.89,0.97);0.8,0.15> | <(0.78,0.81,0.92,0.96),(0.75,0.81,0.92,1);0.65,0.25> |
| C23 | <(0.35,0.41,0.54,0.59),(0.3,0.41,0.54,0.64);0.5,0.5> | <(0.56,0.61,0.74,0.79),(0.53,0.61,0.74,0.84);0.65,0.25> | <(0.56,0.61,0.74,0.79),(0.53,0.61,0.74,0.84);0.65,0.25> | <(0.56,0.61,0.74,0.79),(0.53,0.61,0.74,0.84);0.65,0.25> | <(0.16,0.21,0.38,0.44),(0.12,0.21,0.38,0.47);0.5,0.5> |
| C24 | <(0.45,0.47,0.66,0.73),(0.43,0.47,0.66,0.75);0.65,0.25> | <(0.57,0.6,0.75,0.81),(0.54,0.6,0.75,0.84);0.65,0.25> | <(0.65,0.72,0.81,0.85),(0.6,0.72,0.81,0.91);0.5,0.5> | <(0.45,0.47,0.66,0.73),(0.43,0.47,0.66,0.75);0.65,0.25> | <(0.28,0.35,0.47,0.52),(0.21,0.35,0.47,0.57);0.5,0.5> |
| C25 | <(0.59,0.64,0.75,0.8),(0.55,0.64,0.75,0.83);0.5,0.5> | <(0.25,0.34,0.42,0.47),(0.17,0.34,0.42,0.53);0.35,0.6> | <(0.77,0.81,0.92,0.96),(0.74,0.81,0.92,1);0.65,0.25> | <(0.65,0.7,0.81,0.85),(0.6,0.7,0.81,0.89);0.5,0.5> | <(0.65,0.7,0.81,0.85),(0.6,0.7,0.81,0.89);0.5,0.5> |
| C26 | <(0.15,0.23,0.3,0.36),(0.07,0.23,0.3,0.43);0.35,0.6> | <(0.3,0.35,0.45,0.51),(0.24,0.35,0.45,0.55);0.35,0.6> | <(0.79,0.83,0.93,0.97),(0.77,0.83,0.93,1);0.65,0.25> | <(0.24,0.3,0.38,0.44),(0.17,0.3,0.38,0.5);0.2,0.75> | <(0.58,0.63,0.73,0.77),(0.54,0.63,0.73,0.8);0.5,0.5> |
| C31 | <(0.41,0.44,0.56,0.61),(0.38,0.44,0.56,0.63);0.5,0.5> | <(0.5,0.54,0.63,0.67),(0.46,0.54,0.63,0.7);0.5,0.5> | <(0.07,0.14,0.21,0.27),(0,0.14,0.21,0.33);0.2,0.75> | <(0.43,0.49,0.56,0.59),(0.38,0.49,0.56,0.63);0.5,0.5> | <(0.41,0.44,0.56,0.61),(0.38,0.44,0.56,0.63);0.5,0.5> |
| C32 | <(1,1,1,1),(1,1,1,1);1,0> | <(0.33,0.33,0.33,0.33),(0.33,0.33,0.33,0.33);1,0> | <(0.24,0.24,0.24,0.24),(0.24,0.24,0.24,0.24);1,0> | <(0.22,0.22,0.22,0.22),(0.22,0.22,0.22,0.22);1,0> | <(0.3,0.3,0.3,0.3),(0.3,0.3,0.3,0.3);1,0> |
| C33 | <(0.33,0.43,0.53,0.63),(0.23,0.43,0.53,0.75);0.2,0.75> | <(0.33,0.43,0.53,0.63),(0.23,0.43,0.53,0.75);0.2,0.75> | <(0.17,0.23,0.33,0.43),(0.1,0.23,0.33,0.53);0.2,0.75> | <(0.25,0.33,0.43,0.53),(0.17,0.33,0.43,0.64);0.2,0.75> | <(0.61,0.67,0.77,0.87),(0.53,0.67,0.77,0.97);0.1,0.9> |
| C34 | <(0.53,0.6,0.68,0.75),(0.45,0.6,0.68,0.83);0.2,0.75> | <(0.53,0.6,0.68,0.75),(0.45,0.6,0.68,0.83);0.2,0.75> | <(0.67,0.73,0.8,0.88),(0.6,0.73,0.8,0.95);0.1,0.9> | <(0.59,0.65,0.73,0.8),(0.53,0.65,0.73,0.88);0.1,0.9> | <(0.05,0.07,0.15,0.22),(0,0.07,0.15,0.28);0.5,0.5> |

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
