# Peer review of "Portfolio Optimization of Photovoltaic/Battery Energy Storage/Electric Vehicle Charging Stations with Sustainability Perspective Based on Cumulative Prospect Theory and MOPSO"

_sustainability, doi:10.3390/su12030985_

Round 1

Reviewer 1 Report

This study proposes a hybrid decision framework under trapezoidal intuitionistic fuzzy environment to optimize the PBES portfolio from sustainability perspective. In this method, the cumulative prospect theory is used to evaluate the performance of potential PBES projects and MOPSO is applied to select the optimal portfolio. 14 criteria are identified to evaluate the investment value of PBES. In the proposed decision framework, some relevant methods including trapezoidal intuitionistic fuzzy sets, cumulative prospect theory and MOPSO algorithm are utilized.

The process of the proposed decision framework is well presented and detailed.

The main contributions of this study are sustained by the presented results for a case in China. 10 feasible PBES projects are considered.

The optimization result is a set of Pareto solutions, which means each one of them cannot dominate another. An analysis is conducted to investigate the optimization result in six different scenarios.

Include scenario 7 (which considers the economic, social and environmental objectives) in your analysis and discuss the obtained results. Minor editing errors; see for example:

On the axis of Fig. 8 – left must be Objective 2

Define the acronyms (abbreviations) at first apparition in text; see for example: MCDM

Author Response

Include scenario 7 (which considers the economic, social and environmental objectives) in your analysis and discuss the obtained results. Minor editing errors; see for example:

On the axis of Fig. 8 – left must be Objective 2

Define the acronyms (abbreviations) at first apparition in text; see for example: MCDM

Response: Thanks for your helpful suggestions.

Scenario 7 is actually the case we studied and the results are shown in section 5.2 in detail. As you suggested, we defined scenario 7 as the base scenario and added the analysis in the discussion. In Fig 8, three graphs refer to scenario 1-3 respectively so the left axises are different. To avoid misunderstanding, we numbered three graphs in Fig 8 with (a)-(c). We checked the manuscript and added the definitions of acronyms in the text.

Reviewer 2 Report

Authors have proposed a multi-objective optimization model for photovoltaic/battery energy storage charging stations. A case study in south China was used to validate the effectiveness of the proposed model. Authors have made a significant contribution to identify 14 decision making criteria considering economy, society and environment. One suggestion would be to improve the organization of the paper. The focus of paper is more on methodology and less on results and discussions. It is advised to improve the results and discussion section. 

Author Response

One suggestion would be to improve the organization of the paper. The focus of paper is more on methodology and less on results and discussions. It is advised to improve the results and discussion section. 

Response: Thanks for your helpful suggestion. As you suggested, we added some results and analysis in section 5.